# BioCoref: Benchmarking Biomedical Coreference Resolution with LLMs

## Abstract

Coreference resolution in biomedical texts presents unique challenges due to complex domain-specific terminology, high ambiguity in mention forms, and long-distance dependencies between coreferring expressions. In this work, we present a comprehensive evaluation of generative large language models (LLMs) for coreference resolution in the biomedical domain. Using the CRAFT corpus as our benchmark, we assess the LLMs' performance with four prompting experiments that vary in their use of local, contextual enrichment, and domain-specific cues such as abbreviations and entity dictionaries. We benchmark these approaches against a discriminative span-based encoder, SpanBERT, to compare the efficacy of generative versus discriminative methods. Our results demonstrate that while LLMs exhibit strong surface-level coreference capabilities, especially when supplemented with domain-grounding prompts, their performance remains sensitive to long-range context and mentions ambiguity. Notably, the LLaMA 8B and 17B models show superior precision and F1 scores under entity-augmented prompting, highlighting the potential of lightweight prompt engineering for enhancing LLM utility in biomedical NLP tasks.

## 1 Introduction

Coreference resolution is the process of identifying entities mentioned in text and grouping all mentions that refer to the same underlying entity Liu et al. (2023). In the biomedical domain, coreference resolution is a particularly difficult task as the literature often contain dense, technical language, frequent use of abbreviations, and complex referential expressions that rely on domain-specific background knowledge Lu & Poesio (2021). For instance, resolving a phrase like "the same strain" in a methods section may require linking it back to the "C57BL/6J mice" mentioned several paragraphs earlier, with no intervening repetition or synonyms. Similarly, phrases such as "the compound" may ambiguously refer to any of several chemical entities introduced earlier in experimental descriptions, particularly when multiple drugs or treatments are discussed in parallel. In such cases, surface string similarity offers little guidance; instead, linguistic disambiguation must be informed by contextual and semantic cues.

Adding to the challenge, many biomedical entities share identical surface forms e.g., a gene and its corresponding protein often have the same name or abbreviation which can confuse automated systems. When clustered by identical surface strings, approximately 65% of the coreference clusters in CRAFT corpus Cohen et al. (2017) consist of repeated mentions Li et al. (2022), emphasizing the need for models that can handle referential ambiguity. Moreover, many coreference links span large textual distances, exceeding the effective context window of conventional models Lu & Poesio (2021); Li et al. (2022). These long-range dependencies and requirements for specialized knowledge contribute to the poor generalization of general-domain coreference systems in biomedical contexts.

Given the emergence of increasingly capable large language models (LLMs), a natural question arises: how well can these general purpose models perform coreference resolution in specialized domains like biomedicine, without any task-specific fine-tuning Gan et al. (2024)? LLMs have demonstrated remarkable abilities in complex reasoning and language understanding via prompt-based zero-shot or few-shot learning, often surpassing traditional models in general-domain tasks. This raises the possibility that their extensive pretraining enables them to handle intricate referential structures, even in domain-specific contexts. While biomedical coreference remains a demanding

task, recent advances suggest that with well-designed prompts and minimal scaffolding, LLMs may be more capable than previously assumed.

In this work, we evaluate coreference resolution in biomedical PubMed articles using two contrasting approaches: a span-based model (SpanBERT-Large) Joshi et al. (2020) trained on general-domain data, and several generative LLMs (LLaMA series) Touvron et al. (2023) prompted to resolve coreference without fine-tuning. Our experiments use the CRAFT corpus, a richly annotated biomedical dataset.

The contributions and objectives of this paper are summarized as follows:

- **Benchmarking LLMs** We compare multiple LLaMA models under different prompting strategies: local-only, contextual, abbreviation-aware, and entity-aware against a span-based BERT baseline, reporting performance on the CRAFT corpus.

- **Domain Analysis:** We identify coreference challenges unique to biomedical text, such as identical mention strings and abbreviation ambiguity, and analyze how each model type handles them through qualitative examples and error patterns.

## 2 RELATED WORK

Coreference resolution in biomedical text is a particularly challenging task due to complex domain-specific terminology, high referential ambiguity, and long-range dependencies. Traditional span-based models such as the end-to-end neural coreference models such as SpanBERT Joshi et al. (2020) have demonstrated strong performance in general domains. However, their reliance on limited context windows and the need for supervised fine-tuning limits their applicability in biomedical settings, where coreference often requires broader semantic grounding.

Traditional approaches also include rule-based and statistical systems O'Connor & Heilman (2013); Ng & Cardie (2002); Soon et al. (2001), followed by neural architectures such as the mention-ranking model Clark & Manning (2016) and end-to-end span-ranking networks Lee et al. (2017); Durrett & Klein (2013); Wiseman et al. (2015); Lee et al. (2018). SpanBERT Joshi et al. (2020) further improved coreference resolution by introducing span-centric pretraining objectives, achieving state-of-the-art results on the OntoNotes Dobrovolskii (2021) benchmark. Despite these advances, such models rely heavily on supervised training and domain-specific tuning, limiting their generalizability to out-of-domain settings like biomedical text.

Large language models (LLMs) like OpenAI GPT Radford et al. (2018) and LLaMA have demonstrated strong zero-shot capabilities in various NLP tasks Brown et al. (2020); Touvron et al. (2023), including aspects of coreference. Recent studies evaluated LLMs' abilities on pronoun resolution and Winograd schemas Liu et al. (2024); Wu et al. (2020) for other downstream tasks such as question-answering and query-based span prediction problems. However, few studies have directly assessed LLMs on span-based or noun phrase coreference, particularly in long or technical documents. Most relevant to our work are recent prompting frameworks that use generative models for structured information extraction Xie et al. (2022), though coreference-specific prompting remains underexplored, especially in specialized domains like biomedical literature.

## 3 TASK OVERVIEW

We evaluate four prompt-based strategies for coreference resolution using large language models over CRAFT-formatted biomedical texts. Each document $D$ is split into paragraphs $p_1, \ldots, p_N$, each containing approximately 200 words. Using Stanza parser, we segment the text into sentences and iteratively append them to each paragraph chunk until the 200-word threshold is reached. If the last sentence causes the word count to exceed 200, it is deferred to the next paragraph. The goal is to output the set of detected mentions $M_i$, their corresponding resolutions $A_i$, and a "resolved" version of each paragraph $R_i$, where each $p_i$ is independently rewritten. Formally:

- Let $\text{LLM}_\phi(\cdot)$ denote the output of the LLM with prompt $\phi$.
- Let $M_i$ be the set of coreferent mentions detected in paragraph $p_i$.

- Let $A_i$ be the set of antecedent resolutions for $M_i$.

- Let $R_i$ be the rewritten paragraph $p_i$ with all mentions in $M_i$ resolved using $A_i$.

- The reconstructed document is $\hat{D} = [R_1, \ldots, R_N]$.

The coreference resolution task involves resolving 4 categories: pronouns, definite and indefinite noun phrases, and abbreviations, as illustrated in Table 1. Each is prompted separately for extraction and resolving.

| Coreference Type | Example Expressions |
|---|---|
| **Pronouns** | *it, they, this, these, those, its, their* |
| **Definite noun phrases** | *the gene, these proteins, such results* |
| **Indefinite noun phrases** | *a protein, some genes, one of the enzymes* |
| **Abbreviation coreference** | *IOP → intraocular pressure* |

Table 1: Coreference categories and example expressions used in our experiments.

To evaluate how different categories of auxiliary information affect coreference resolution, we design four prompting configurations: (1) a local-only setup with no external context, (2) a reference-based setup that incorporates the first paragraph as a fixed disambiguation source, (3) an abbreviation-aware setup using a dictionary of extracted abbreviation-definition pairs, and (4) an entity-aware setup using a list of biomedical entities extracted from the document. Algorithm 1 summarizes these 4 styles of the prompting experiments.

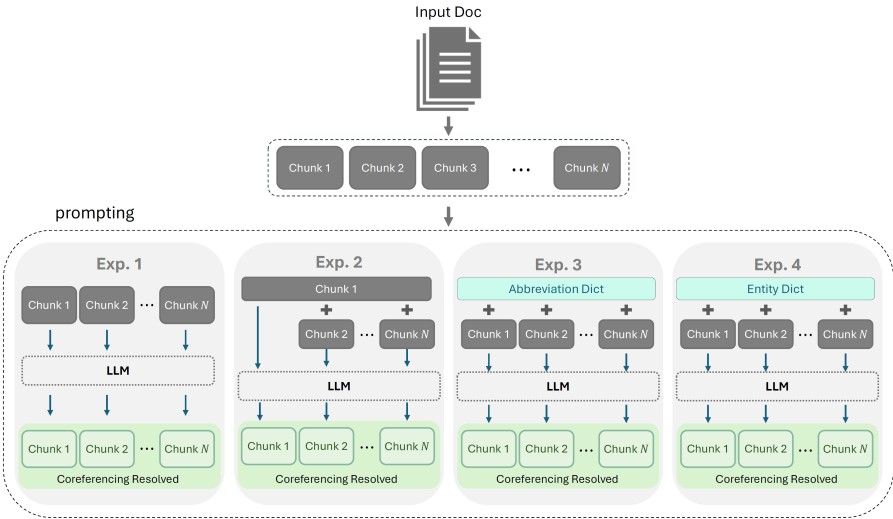

Figure 1: Overview of the coreference resolution pipeline under four prompting strategies. Each chunk is processed by an LLM independently (Exp. 1), with prior context (Exp. 2), or with auxiliary inputs such as abbreviation (Exp. 3) or entity dictionaries (Exp. 4).

## 3.1 EXPERIMENT 1: LOCAL-ONLY RESOLUTION (BASELINE)

$$R_i = \text{LLM}_{\text{local}}(p_i)$$

In this initial experiment, we investigate the effectiveness of local coreference resolution by prompting LLMs to resolve coreference chains within short, isolated 200-word segments of a biomedical article. Each chunk is independently passed to the LLM. The goal is to assess how well the model performs coreference resolution without any cross-paragraph or global context.

This design reflects a naïve but computationally inexpensive strategy: it minimizes prompt complexity and token limits, while simulating how local context alone may or may not suffice for resolving

---

**Algorithm 1** Prompt-Based Coreference Resolution

---

**Require:** Document $D$, ExperimentType $\in$ {LOCAL, REF_CTX, ABBR, ENTITY}, Model LLM
**Ensure:** ResolvedDocument $\hat{D}$, MentionSets $\mathcal{M}$, ResolutionSets $\mathcal{A}$
  1: Split $D$ into paragraphs: $[p_1, p_2, \ldots, p_N]$
  2: Initialize auxiliary content $C \leftarrow \emptyset$
  3: **if** ExperimentType = ABBR **or** ExperimentType = ENTITY **then**
  4:     $C \leftarrow$ EXTRACTCONTEXTINFO($D$, type=ExperimentType)
  5: **end if**
  6: Initialize $\hat{D} \leftarrow [\ ]$, $\mathcal{M} \leftarrow [\ ]$, $\mathcal{A} \leftarrow [\ ]$
  7: **for** $i = 1$ to $N$ **do**
  8:     $p \leftarrow p_i$
  9:     **if** ExperimentType = REF_CTX **then**
10:       reference $\leftarrow p_1$ {Use Paragraph 1 as fixed reference}
11:     **else**
12:       reference $\leftarrow \emptyset$
13:     **end if**
14:     prompt $\leftarrow$ BUILDPROMPT(reference, $p$, $C$, ExperimentType)
15:     response $\leftarrow$ QUERYLLM(LLM, prompt)
16:     result $\leftarrow$ PARSEJSON(response)
17:     $\hat{D}$.append(result["Rewritten_Paragraph"])
18:     $\mathcal{M}$.append(result["Extracted_Expressions"])
19:     $\mathcal{A}$.append(result["Resolutions"])
20: **end for**
21: **return** $\hat{D}$, $\mathcal{M}$, $\mathcal{A}$

---

biomedical coreference phenomena. We made a separate inference run for the 4 coreferencing categories.

This framework allows us to isolate and quantify the limitations of local-only resolution in biomedical texts It also establishes a baseline against which we can measure subsequent experiments incorporating other resources, such as abbreviation expansion (Experiment 3).

### 3.2 EXPERIMENT 2: COREFERENCE RESOLUTION WITH LOCAL AND REFERENCE CONTEXT

$$R_i = \text{LLM}_{\text{ref}}(p_1, p_i)$$

- **Prompt:** Provide $p_1$ and $p_i$, instructing the LLM to use the former to disambiguate references in the latter.

- **Purpose:** Test the incremental benefit of a reference paragraph for resolving inter-sentential and cross-paragraph coreferring mentions, without overloading the prompt size.

Building upon the limitations identified in Experiment 1, where coreference resolution was performed in isolation within 200-word chunks, we introduce an additional layer of local context to guide the LLM. In this experiment, each prompt to the model includes not only the target paragraph, but also the first 200-word paragraph in the paper, which carries most of the referential information introduced in the paper and can therefore act as an answer key for the unresolved references in the target paragraph.

Each prompt is structured with two segments: a reference block (Paragraph 1) and a focus block (Paragraph $n$), with explicit instructions for the LLM to resolve all ambiguous mentions in the focus block using context from the reference. This experiment assesses the impact of lightweight contextual bridging on coreference resolution quality. Compared to the purely local setting in Experiment 1, this approach tests whether even a single paragraph of surrounding context can significantly improve the coherence and referential clarity of LLM-generated outputs, without exceeding typical token limits or requiring full-document inputs.

### 3.3 EXPERIMENT 3: ABBREVIATION-AWARE COREFERENCE RESOLUTION USING LLM-EXTRACTED DICTIONARIES

Let $A = \{(a_j, \alpha_j)\}$ be abbreviation-definition pairs extracted from the first 750 words using the GPT-4o.

$$R_i = \text{LLM}_{\text{abbr}}(A; p_i)$$

- **Prompt:** "Here is a list of abbreviations A. the model is requested to extract all the coreferencing categories in separate runs, then, rewrite paragraph $p_i$ by expanding ambiguous abbreviations and resolving references."

- **Purpose:** Leverage explicit abbreviation knowledge to aid disambiguation of biological mentions.

Biomedical texts frequently employ abbreviations for complex names, which can cause substantial ambiguity in coreference resolution. In this experiment, we assess whether providing LLMs with a structured abbreviation dictionary improves coreference resolution compared to unstructured context, such as the reference paragraphs used in Experiment 2. To build this dictionary, we parse the first 750 wordsof each document using Stanza and extract abbreviation-definition pairs (e.g., `APP` = "amyloid precursor protein") using the GPT-4o interface. These pairs are then validated against the CRAFT corpus to ensure correctness. The resulting Abbreviation List is passed as auxiliary input during prompting.

### 3.4 EXPERIMENT 4: ENTITY-AWARE COREFERENCE RESOLUTION USING LLM-EXTRACTED DICTIONARIES

Let $E = \{e_k\}$ be key biomedical entities extracted from the first 750 words using GPT4o.

$$R_i = \text{LLM}_{\text{entity}}(E; p_i)$$

- **Prompt:** "Here is a list of detected biomedical entities $E$. Extract all the coreferencing mentions, then, rewrite paragraph pi by expanding ambiguous abbreviations and resolving references"

- **Purpose:** Provide broader semantic grounding than abbreviations alone, to evaluate whether entity awareness supports coherent coreference resolution.

In this experiment, we examine whether incorporating explicit biomedical entity information into the prompting process can enhance the performance of large language models on coreference resolution. Instead of only relying on implicit context or abbreviation mappings, we provide the LLM with a curated Entity List; a list of biomedical terms extracted using GPT-4o interface from the first 750 words of each document and validated against the CRAFT corpus to ensure correctness.

This entity list serves as a form of semantic grounding. For each paragraph in the input article, the LLM is prompted with both the paragraph and the corresponding entity list. The model is then asked to resolve any ambiguous mentions by aligning them with the most probable entry in the entity list and rewriting the paragraph accordingly.

## 4 DATASET

The Colorado Richly Annotated Full-Text (CRAFT) corpus is an annotated biomedical dataset consisting of 67 full-text, open-access journal articles from PubMed Central. The dataset includes extensive manual annotations for biomedical concepts, syntactic structure, and coreference identity chains. In Version 2.0, CRAFT introduced comprehensive coreference annotations, comprising nearly 30,000 coreference relations that span both identity and appositive links across base noun

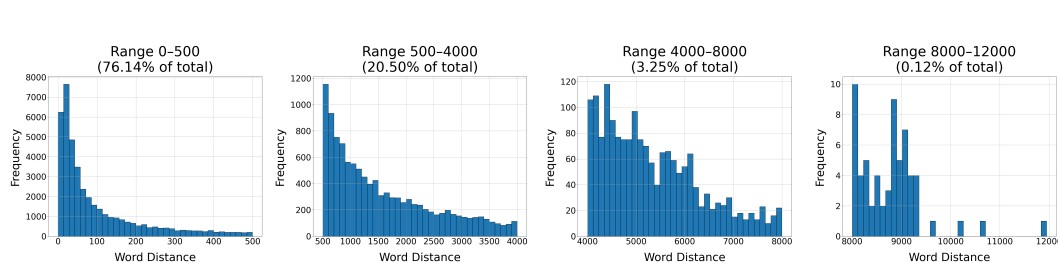

Figure 2: Distribution of word distances between coreferent mentions in biomedical texts, grouped into four ranges.

phrases[1]. We tested the language models on 50 articles selected at random, which are available on the paper's GitHub[2].

Figure 2 shows the distribution of word-level distances between entities and their coreferential mentions in the CRAFT dataset, segmented into four ranges. While most coreference links (76.14%) fall within 0–500 words, a substantial portion (over 23%) spans distances up to 12,000 words. These long-range dependencies present a significant challenge for coreference models, which must retain and retrieve relevant context across spans exceeding the effective window of standard neural architectures.

In addition to these distance-based challenges, the dataset itself contains numerous subtle forms of annotations variability that could be challenging for LLMs. For example, mentions of age: "3 months of age"; statistical and symbolic variables may be abbreviated as single letters like "n"; temporal references: "1997"; dosage information often mixes numbers and units, as in "9 mg/kg xylazine"; and biological sex descriptors appear inconsistently as "Male," "M," "Female," or "F." These heterogeneous expression patterns increase the complexity of accurate entity recognition and coreference resolution, making the dataset a very helpful benchmark for the current LLMs.

## 5 MODELS

For our span-based baseline, we evaluate **SpanBERT-large-cased** model Joshi et al. (2020), a span-optimized transformer pretrained on masked span prediction, on one experiment of coreferencing resolution. Input documents are segmented into 150-word chunks using `Stanza` Qi et al. (2020), as the model counting handle larger context inputs. We normalized the document chunking sizes using stanza for all the experiments to assure the same chunk indices production for proper evaluation. After chunking the document, noun/pronoun mentions are extracted via `spaCy` Vasiliev (2020). Each mention is encoded using SpanBERT's final-layer embeddings and clustered via clustering with cosine similarity ($\tau = 0.4$) to group together mentions that semantically refer to the same entity, approximating coreference. For the generative approach, we evaluate three open-weight LLaMA models on each of the 4 coreferencing experiments:

- **LLaMA 3.3 70B-Instruct** Meta AI (2024a): a high-capacity model (128k context) released in April 2024.

- **Llama-3.1-8B-Instruct** Meta AI (2024b): a compact model optimized for efficient text-only inference.

- **LLaMA 4 Scout 17B** Meta AI (2025): a 2025 multimodal model with a 10M-token context window and Mixture-of-Experts architecture.

---

[1] https://github.com/UCDenver-ccp/CRAFT
[2] https://github.com/XXXX/BioCoref

## 6 RESULTS ANALYSIS

To ensure accurate evaluation, we removed 9335 gold-standard annotations from the CRAFT selected article that were not connected via relation entries. We then matched predicted resolutions against the remaining 83,608 annotated spans using partial character overlap ($\geq 2$ characters), ensuring case-insensitive alignment. Predictions were extracted from structured JSON when available, or via a fallback regex parser. Precision, recall, and F1 scores were computed at the mention level, with unmatched predictions treated as false positives and missed gold spans as false negatives.

First, we report the performance of the span-based baseline **SpanBERT-large**, which achieves an F1 score of only 0.1322. This highlights the difficulty of biomedical coreference resolution for traditional models, due to limited context windows, domain-specific terminology, and the mismatch between general-domain fine-tuning and specialized biomedical discourse.

All open-weight LLM experiments were conducted on a high-performance Google Cloud VM instance of type `a2-highgpu-8g`, equipped with 96 vCPUs, 680 GB of RAM, and 8 NVIDIA A100 GPUs (40 GB each). The environment was provisioned with the `c0-deeplearning-common-cu118-v20241118-debian-11-py310` boot disk image, ensuring compatibility with CUDA 11.8 and PyTorch 2.x frameworks.

Table 2: Performance metrics for `LOCAL` and `REF_CTX` tasks

| Model | LOCAL | | | REF_CTX | | |
|---|---|---|---|---|---|---|
| | **Precision** | **Recall** | **F1 Score** | **Precision** | **Recall** | **F1 Score** |
| LLaMA 70B | 0.800 | 0.458 | 0.583 | 0.805 | 0.390 | 0.525 |
| LLaMA 17B | 0.825 | 0.613 | 0.704 | 0.850 | **0.573** | **0.685** |
| LLaMA 8B | **0.874** | **0.723** | **0.791** | **0.906** | 0.539 | 0.676 |

Table 3: Performance metrics for `abb_dictionary` and `entity_dictionary` tasks

| Model | ABBR | | | ENTITY | | |
|---|---|---|---|---|---|---|
| | **Precision** | **Recall** | **F1 Score** | **Precision** | **Recall** | **F1 Score** |
| LLaMA 70B | 0.844 | 0.395 | 0.538 | 0.826 | 0.379 | 0.519 |
| LLaMA 17B | **0.919** | 0.400 | 0.558 | **0.891** | **0.633** | **0.740** |
| LLaMA 8B | 0.868 | **0.653** | **0.745** | 0.882 | 0.551 | 0.678 |

Our results reveal consistent patterns in how generative LLMs perform on coreference resolution in the biomedical domain:

**Model Scale vs. Effectiveness.** LLaMA 8B and the 17B models outperformed the 70B variant across all experiments. This suggests that model scale alone is not indicative of better coreference performance, especially in domain-specific tasks. One likely explanation is that smaller models generalize more conservatively and make fewer overconfident errors, whereas larger models despite stronger generative capacity may be more susceptible to prompt misalignment and semantic overreach.

To further probe local-only performance, we ran an additional variant of Experiment 1 where paragraphs were selected not sequentially by size of 200-word window, but based on the most frequent reference distance observed in the CRAFT corpus, which is 500 words. The goal of this experiment is to test whether LLMs perform better when the input chunk maximally aligns with natural coreference distances, rather than strict 200-word segmentation. Results are shown in Table 4.

**Impact of Coreference Distance.** A noticeable drop in recall and F1 for LLaMA 17B and 8B is showin in this distance-aware local experiment variant. This suggests that proximity alone is insufficient for robust coreference; many biomedical entities require contextual cues beyond sentence-local

Table 4: Performance metrics for distance-aware local coreference resolution

| Model | Precision | Recall | F1 Score |
|---|---|---|---|
| LLaMA 70B | 0.794 | 0.430 | 0.557 |
| LLaMA 17B | 0.841 | 0.554 | 0.668 |
| LLaMA 8B | **0.892** | **0.601** | **0.718** |

information. The LLaMA 8B model still achieved the highest F1 score. However, the decline in the performance, as the context size increases is now proven by this experiment as well as (experiment 2: REF_CTX).

**Reference Context Has Mixed Effects.** In Experiment 2, which incorporated a fixed reference paragraph to aid disambiguation, recall often dropped compared to the purely local setup (Experiment 1), particularly for LLaMA 8B and 70B. This result suggests that unstructured introduction to the entities does not necessarily guide the model to better locate the referencing in later paragraphs in the paper.

**Structured Dictionaries Improve Recall.** Both the abbreviation-aware and entity-aware settings (Experiments 3 and 4) demonstrated measurable gains in recall and F1, especially for the 8B model. When supplied with structured input, such as abbreviation definitions or pre-extracted entity lists, the models were better able to identify correct antecedents. This effect was most pronounced in LLaMA 8B. These findings are consistent with evidence that structured, grounded prompting improves performance in information extraction tasks.

Overall, these findings highlight both the promise and current limitations of generative LLMs for biomedical coreference. While auxiliary signals such as abbreviation and entity dictionaries can meaningfully boost performance, LLMs still struggle to integrate multi-paragraph context and resolve less explicit coreferences.

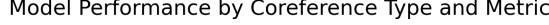

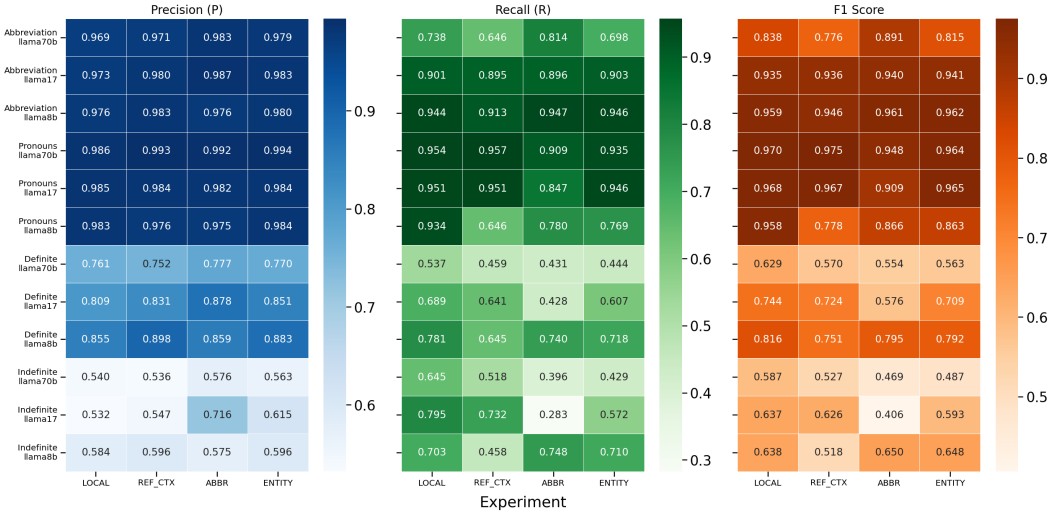

Figure 3: Heatmap of precision, recall, and F1 scores for LLaMA models (70B, 17B, 8B) across four experimental setups (LOCAL, REF CTX, ABBR, ENTITY) and coreference categories (pronouns, indefinite NPs, abbreviations, definite NPs).

**Coreference Type Sensitivity and Model Behavior.** To better understand how the LLMs handle different forms of coreference, we evaluated their performance across the four categories: pronouns,

indefinite noun phrases, abbreviations, and definite noun phrases, under the four contextual setups: `LOCAL`, `REF_CTX`, `ABBR`, and `ENTITY`.

To support this analysis, we developed a post-processing pipeline to classify the predicted and the CRAFT ground truth mentions into four types using lexical heuristics, as the CRAFT dataset does not label coreference types. Pronoun, indefinite, and abbreviations dictionaries we prepared from the source articles and are available on our GitHub. Remaining mentions were treated as definite noun phrases. Each was then evaluated using the model's original prediction labels to compute precision, recall, and F1 scores per type.

As shown in Figure 3, pronoun coreference consistently achieved the highest F1 scores across all LLaMA models, with LLaMA 70B reaching 0.975 under the `REF_CTX` setup. This strong performance is likely due to the frequent occurrence of pronouns in pretraining corpora and their reliance on short-range syntactic cues. Complementary evidence from Figure 4 confirms that pronouns were also resolved in high absolute counts across experiments, especially by LLaMA 17B under minimal context.

Abbreviation coreference also exhibited strong performance, particularly under the `ABBR` and `ENTITY` experiments. Injecting abbreviation dictionaries yielded a noticeable increase in both F1 scores (e.g., LLaMA 8B achieving 0.961 in `ABBR`) and mention resolution counts. These results affirm that domain-specific cues can significantly enhance model understanding of biomedical abbreviation references.

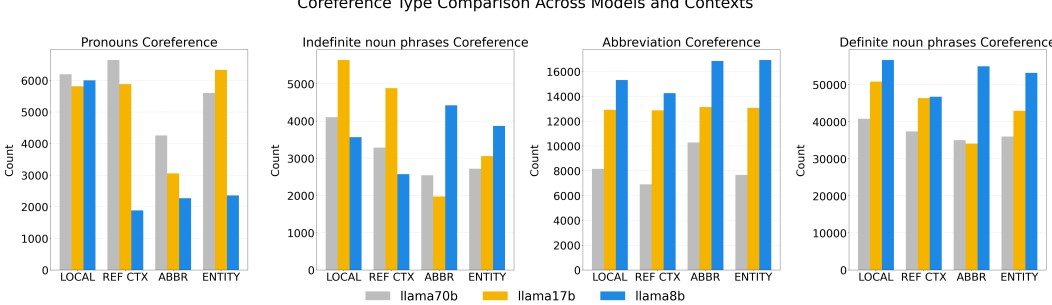

Figure 4: Extracted coreference type counts by model and context.

## 7 CONCLUSION

Our study presented a systematic evaluation of generative large language models (LLMs) for coreference resolution in the biomedical domain. We benchmarked three LLaMA models across four prompt-based settings and compared them to a span-based baseline, using the richly annotated CRAFT corpus for evaluation.

Overall, these results highlight the relationship between model size, coreference category, and the design of contextual input. They emphasize that targeted domain-specific augmentation, such as structured dictionaries, can have a greater impact on performance than model scale alone. Notably, smaller models can match or even exceed the performance of larger ones when paired with carefully designed prompts. Future directions should explore fine-tuning strategies, integration of external biomedical knowledge, and hybrid generative extractive systems to further enhance recall and robustness.

ACKNOWLEDGMENTS

All Open-weight LLMs discussed in the paper were used for the purpose of inference and evaluation. OpenAI Models were used to refine approximately half of the manuscript, focusing solely on enhancing readability and grammatical accuracy. Great care was taken to ensure that no new content was introduced or that existing ideas were altered during this process.

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

## A   APPENDIX

### A.1   PROMPT TEMPLATE

To guide the language model's behavior consistently across experiments, we employ a structured system prompt for each coreference type, and this prompt instructs the model to identify and resolve only a targeted subset of coreference expressions. In this example, which is a portion of the prompt, the focus is on definite noun phrase coreferences within a paragraph, while explicitly excluding pronouns, indefinite expressions, and abbreviations.

**System Prompt**

You are a scientific language model with expert-level understanding of coreference resolution.
Your task is to extract and resolve ONLY **definite noun phrase coreferences** (e.g., "the gene", "these proteins", "such results") within the paragraph.
**Skip** the following:

- Pronouns (e.g., "it", "they")
- Indefinite noun phrases (e.g., "a result", "some proteins")
- Abbreviations (e.g., "IOP")

Follow these steps:

1. Extract coreference expressions that appear *verbatim* in the paragraph. **Do NOT invent or rephrase them**.
2. For each expression, resolve it to its correct antecedent using context from the same paragraph.
3. Rewrite the paragraph by substituting each extracted expression with its resolved referent.

**DO NOT** paraphrase, summarize, add, remove, or reorder any content. **Preserve the original wording and sentence structure, except for the substitutions.**

**Expected JSON Output Schema:**

```
{
  "Extracted_Expressions": [
    "[expression_1]",
    "[expression_2]"
  ],
  "Resolutions": {
    "[expression_1]": "[detailed explanation describing the antecedent]",
    "[expression_2]": "[detailed explanation describing the antecedent]"
  },
  "Rewritten_Paragraph": "[the rewritten paragraph, identical except for
      substitutions]"
}
```

**Example:**

*Input:* "These results were unexpected. They indicate a new trend."
*Rewritten:* "The results were unexpected. The results indicate a new trend."

**Example Output:**

```
{
  "Coreference_Resolution": {
    "Extracted_Expressions": [
      "IOP",
      "IOPs",
      "They"
    ],
    "Resolutions": {
      "IOP": "intraocular pressure",
      "IOPs": "intraocular pressures",
      "They": "Genetically distinct mouse strains"
    },
    "Rewritten_Paragraph": "Intraocular pressure in genetically distinct
        mice: an update and strain survey..."
  }
}
```

