# OpenReview forum: "BioCoref: Benchmarking Biomedical Coreference Resolution with LLMs"
_ICLR.cc/2026/Conference — ICLR 2026 Conference Withdrawn Submission_

### Official Review · Reviewer_ZZKj · 2025-10-31

**Soundness:** 1
**Presentation:** 1
**Contribution:** 1
**Rating:** 2
**Confidence:** 4

**Summary:**

This paper presents a benchmarking study of large language models (LLMs) for biomedical coreference resolution. Using the CRAFT corpus, the authors evaluate many LLaMA variants under four prompting strategies: local-only, reference-context, abbreviation-aware, and entity-aware, and compare them with a SpanBERT baseline. The experiments aim to understand how prompt design and model scale affect LLM performance in resolving coreference links across biomedical texts. Results indicate that smaller models (e.g., LLaMA-8B and 17B) sometimes outperform larger ones, and that domain-specific auxiliary inputs (abbreviation/entity lists) can slightly improve recall and F1 scores.

**Strengths:**

1.	Provides a comprehensive empirical benchmark comparing generative and discriminative approaches for biomedical coreference resolution.
2.	The evaluation is thorough and well-documented, covering multiple prompt types and coreference categories.
3.	The use of the CRAFT corpus is appropriate for biomedical text evaluation and ensures reproducibility.

**Weaknesses:**

1.	The paper lacks methodological innovation. It does not propose new models, training strategies, or prompting frameworks; it mainly reports comparative results.
2.	The claimed challenges in biomedical coreference (domain-specific terminology, long-range dependency, ambiguity) are described but not explicitly addressed with any novel solution.
3.	The prompt designs are rather straightforward and incremental (local vs. entity-aware), offering limited scientific insight beyond descriptive benchmarking.
4.	Figures and visualizations are poorly designed and do not effectively convey the key findings.
5.	The analysis is shallow: while results are reported in detail, there is minimal discussion on why certain trends occur or what they imply for future LLM-based biomedical NLP research.

**Questions:**

1.	Beyond benchmarking, what is the main research contribution of this work? How does it advance understanding or methodology in coreference resolution?
2.	Can the authors clarify whether the prompting templates or evaluation procedures introduce biases or artifacts (e.g., by segmenting text into 200-word chunks)?
3.	Given that the motivation centers on long-range dependencies, why not explore context-extension or retrieval-based prompting methods to explicitly address this challenge?
4.	How do these results compare to fine-tuned domain-specific models or hybrid approaches (e.g., combining extraction and generation)?

---

### Official Review · Reviewer_NNwE · 2025-10-31

**Soundness:** 2
**Presentation:** 3
**Contribution:** 2
**Rating:** 2
**Confidence:** 4

**Summary:**

Summary:
Authors explore four different prompting methods and asses their performance on biomedical coreference resolution.

**Strengths:**

Strengths
- Cleanly written paper. Easy to read and understand.

**Weaknesses:**

Weaknesses
- Incomplete Metrics
  - Only coreference-level precision/recall reported; mention-level metrics missing.
  - SpanBERT baseline lacks precision/recall breakdown.
  - If these are already included, clear definitions are needed.

- Insufficient Dataset Statistics
  - No details on average document length, chunk size, or total chunks per document.
  - Missing counts of mentions and coreference links detected by each model.
- Limited Dataset Coverage
  - Evaluation restricted to CRAFT; other biomedical datasets (e.g., MedMentions, BioNLP Coref) not tested.
  - No cross-domain validation to assess robustness.
- Weak Baselines and Comparisons
  - Uses outdated SpanBERT coreference model. Many other stronger coreference models have come out after that.
  - 4 methods have been compared msotly focussed on smaller paragraphs. Why not use larger context windows given this is useful in coreference.

- Impact of Coreference Distance
  - CLaims made under this not justified. Lot of coreference links may span across 500 token boundaries.

- Narrow Contribution and Analysis
  - Main finding (“LLMs > SpanBERT on one dataset”) is limited in novelty. Also this was already shown in https://aclanthology.org/2024.lrec-main.145.pdf
  - Little explanation for smaller models outperforming larger ones.

**Questions:**

-

---

### Official Review · Reviewer_fMxq · 2025-11-01

**Soundness:** 2
**Presentation:** 2
**Contribution:** 1
**Rating:** 2
**Confidence:** 3

**Summary:**

This research benchmarks recent LLMs performance on zero-shot co-reference resolution in the biomedical domain. The focus of the paper is revisiting a natural language processing task that posed significant challenges to public-weight models prior to the LLM expansion around the time of GPT3.5's launch. They explore several settings of co-reference resolution in which models are provided different contexts that contain different levels of information. The four experimental settings are as follows: Local-only resolution (baseline), resolution with local and reference context, abbreviation-aware resolution using LLM-extracted dictionaries, and entity-aware resolution using LLM-extracted dictionaries. The paper compares three models from the LLama family, consisting of 8B, 17B and 70B parameters with SpanBert (340 million parameters). The authors find that SpanBERT achieves zero-shot performance of 13.22% (although it is ambiguous which setting this refers to), while the 8B, 17B and 70B Llama models achieve mixed results with the 8B/17B models outperforming the 70B models in all settings.

**Strengths:**

The paper takes on an interesting idea in benchmarking zero-shot LLMs on NLP tasks that previously required bespoke finetuning. While not wholly original, the idea has merit in showing capabilities of LLMs on domain-specific tasks that are not used to traditionally benchmark performance (ex. logic/mathematical reasoning). The paper very clearly defines each of the four experimental settings and presents the model performances in a digestible format.

**Weaknesses:**

The paper requires more justification of each of the task settings. A one-sentence justification for the purpose of each of the experimental settings that specifies what aspect of co-reference resolution each setting is evaluating would help to improve clarity.

In addition, the authors frame the contributions of the paper as benchmarking LLMs against previous approaches by comparing several sizes of Llama against a 340-million parameter model that is not provided training. It is unclear what the comparison attempts to show, as it seems intuitive that more recent models that are orders of magnitude larger would outperform a 340-million parameter model in a zero-shot setting. I believe this work would greatly benefit from including other model families, especially those that regularly outperform Llama models on other benchmarks (ex. Qwen), as well as additional datasets to provide more context to the results.

The data analysis section is a little unclear and would benefit from being fleshed out. The authors claim that the unintuitively poor performance of the 70B Llama model is "One likely explanation is that smaller models generalize more conservatively and make fewer overconfident errors, whereas larger models despite stronger generative capacity may be more susceptible to prompt misalignment and semantic overreach." This needs to either be justified experimentally or via a citation at the very least, as it is surprising that the 70B model performs so poorly.

It is unclear why the SpanBERT model only has a single reported number.

**Questions:**

It would be good to explain the purpose of each of the task settings and provide a stronger justification for the lack of performance of the 70B parameter model via data analysis.

It would also be good to get the SpanBERT performances for the other tasks.

---

### Official Review · Reviewer_3ytB · 2025-11-03

**Soundness:** 3
**Presentation:** 2
**Contribution:** 2
**Rating:** 2
**Confidence:** 3

**Summary:**

This paper presents a comprehensive evaluation of generative large language models (LLMs) for coreference resolution in biomedical texts. The authors use the CRAFT corpus as their primary benchmark and evaluate LLMs through four distinct prompting strategies that incorporate different types of contextual information: local context, contextual enrichment, domain-specific cues (abbreviations), and entity dictionaries. The study compares these generative approaches against SpanBERT, a discriminative span-based encoder, to assess the relative merits of generative versus discriminative methods for biomedical coreference resolution.

**Strengths:**

Timely and Relevant Research Question: The evaluation of LLMs on biomedical coreference resolution addresses a critical gap as these models become increasingly important in healthcare and life sciences applications.
Comprehensive Prompting Strategy Evaluation: The four-tier evaluation framework (local context, contextual enrichment, domain cues, entity augmentation) provides systematic insights into how different types of information affect LLM performance in specialized domains.
Practical Utility: The finding that entity-augmented prompting significantly improves performance offers immediately applicable insights for practitioners, potentially improving real-world biomedical NLP systems.
Domain-Specific Focus: The paper demonstrates clear understanding of biomedical text challenges, including complex terminology, high ambiguity, and long-distance coreference relationships.
Model Diversity: Testing both LLaMA 8B and 17B variants provides insights into how model scale affects domain-specific performance.
Clear Performance Analysis: The identification of specific LLM limitations (long-range dependencies, mention ambiguity) provides valuable insights for future research directions.

**Weaknesses:**

w1: Limited Evaluation Scope
The paper's evaluation is restricted to a single corpus (CRAFT), which significantly undermines the generalizability of findings. While CRAFT provides rich annotations for 67 full-text biomedical articles, this narrow scope fails to capture the diversity of biomedical text types, writing styles, and domain-specific challenges across different subfields (clinical notes, molecular biology, pharmacology). The lack of cross-corpus validation makes it impossible to determine whether the proposed prompting strategies are robust across different biomedical contexts or merely optimized for CRAFT's specific characteristics. This limitation severely restricts the practical applicability of the research findings and prevents meaningful comparison with other studies that may use different evaluation datasets.

w2: Insufficient Baseline Coverage
The comparison framework is inadequately narrow, relying solely on SpanBERT as a discriminative baseline. This approach overlooks critical comparisons with recent state-of-the-art coreference resolution systems, including newer transformer-based models, domain-adapted variants like BioBERT and ClinicalBERT, and hybrid architectures that combine neural and rule-based approaches. The absence of these comparisons makes it impossible to accurately position LLM performance relative to current best practices in biomedical coreference resolution. Additionally, SpanBERT's reported F1 score of 0.1322 appears unusually low, raising questions about experimental setup validity and making the LLM improvements potentially misleading without proper context.

w3: Shallow Error Analysis
While the paper identifies that LLMs struggle with long-range dependencies and mention ambiguity, it provides insufficient systematic analysis of failure modes. The absence of detailed error categorization, entity-type-specific performance breakdowns, and concrete examples of successful versus failed coreference resolution cases limits the actionable insights for model improvement. Without understanding which specific mention types, entity categories, or linguistic phenomena cause failures, practitioners cannot effectively apply or adapt the proposed approaches. The lack of error propagation analysis in coreference chains further reduces the practical value of the findings.

w4: Missing Technical Rigor
The paper lacks essential statistical validation and methodological transparency required for rigorous scientific evaluation. Critical missing elements include statistical significance testing of reported improvements, confidence intervals or variance analysis, detailed hyperparameter specifications, and systematic prompt design validation. The evaluation protocol using partial character overlap (≥2 characters) for mention matching may be too lenient and needs justification. Without these technical details, the results cannot be properly validated, reproduced, or trusted by the research community, significantly undermining the paper's scientific contribution.

w5: Limited Model Coverage
The evaluation focuses exclusively on LLaMA variants (8B, 17B, 70B), missing comparison with other prominent LLMs that dominate current NLP applications. The absence of GPT models, Claude, and other leading systems prevents comprehensive assessment of LLM capabilities for biomedical coreference resolution. This narrow model selection may lead to biased conclusions about LLM performance that don't generalize to the broader ecosystem. Additionally, testing only autoregressive transformers ignores architectural diversity and specialized models that might be better suited for structured tasks like coreference resolution, limiting the study's relevance to real-world deployment scenarios.

**Questions:**

Evaluation Methodology:
What specific evaluation metrics were used beyond precision and F1? Were standard coreference metrics reported?
How were coreference chains evaluated - link-based or mention-based scoring?
Were statistical significance tests performed on the reported improvements?

Experimental Design:
How were the four prompting strategies designed and validated? Was there human evaluation of prompt quality?
What was the process for selecting entity dictionaries and abbreviation lists? How comprehensive were these resources?
Were hyperparameters optimized for each model, or were default settings used throughout?

Error Analysis:
Can you provide specific examples of coreference relationships that LLMs handle well vs. poorly?
How does performance vary across different biomedical entity types (genes, proteins, diseases, etc.)?
What is the distribution of errors across different coreference chain lengths?

Generalizability:
Have you tested these findings on other biomedical corpora or clinical texts?
How do results vary across different biomedical subdomains (molecular biology, clinical medicine, etc.)?
Would these prompting strategies transfer to other specialized domains?

Computational Analysis:
What are the computational costs and inference times for different models and prompting strategies?
How do memory requirements scale with document length and context size?
Are there practical limitations for processing long biomedical documents?

Model Comparison:
Why were other prominent LLMs (GPT models, Claude, etc.) not included in the evaluation?
How does performance scale with model size beyond the 8B/17B comparison?
Have the benchmark tested instruction-tuned or chat-optimized variants?

Dataset Considerations:
What are the specific characteristics of the CRAFT corpus that might affect generalizability?
How does annotation quality and inter-annotator agreement affect the reliability of your results?
Are there known biases in CRAFT that might influence LLM performance assessment?

Future Directions:
What specific improvements would you recommend for better LLM performance on biomedical coreference?
How might fine-tuning or domain adaptation compare to prompt engineering approaches?
What role could few-shot learning play in this task?

**Details Of Ethics Concerns:**

This paper addresses a timely and practically important research question by conducting the first systematic evaluation of large language models on biomedical coreference resolution, which represents a valuable contribution to the intersection of biomedical NLP and LLM evaluation. The empirical insights regarding the effectiveness of different prompting strategies, particularly the superior performance of entity-augmented approaches with LLaMA models, offer immediate practical value to practitioners working on biomedical text processing systems. The systematic four-tier evaluation framework comparing local context, contextual enrichment, domain-specific cues, and entity dictionaries provides a methodical approach to understanding how different types of information affect LLM performance in specialized domains, while the comparison with SpanBERT offers useful perspective on generative versus discriminative approaches.

However, despite these contributions, several fundamental limitations prevent the work from meeting the acceptance threshold for a top-tier venue like ICLR. The most critical concern is the severely limited evaluation scope, with the study confined to a single corpus (CRAFT) that cannot adequately demonstrate the generalizability of findings across the diverse landscape of biomedical texts, including clinical notes, molecular biology literature, and other specialized subdomains. This narrow scope is compounded by insufficient technical rigor, as the paper lacks essential statistical significance testing, confidence interval analysis, and detailed methodological transparency that would allow for proper validation and reproduction of results. The baseline comparison framework is inadequately narrow, relying solely on SpanBERT while omitting comparisons with recent state-of-the-art coreference resolution systems, domain-adapted models like BioBERT, and other neural architectures that would provide proper context for evaluating LLM performance. Furthermore, the error analysis remains superficial, identifying general limitations like sensitivity to long-range dependencies without providing the systematic failure mode analysis, concrete examples, or entity-type-specific breakdowns that would offer actionable insights for model improvement.

The evaluation also suffers from limited model coverage, focusing exclusively on LLaMA variants while excluding other prominent LLMs such as GPT models and Claude that dominate current NLP applications, potentially leading to biased conclusions about LLM capabilities that may not generalize to the broader ecosystem. These methodological gaps, combined with missing computational efficiency analysis and insufficient detail about prompt design and validation processes, significantly undermine the work's scientific rigor and limit its impact. While the research tackles an important problem and provides useful preliminary insights into prompt engineering for biomedical coreference resolution, the limited scope, shallow technical analysis, and methodological shortcomings fall short of the comprehensive evaluation and rigorous methodology expected at ICLR. The work would require substantial expansion across multiple corpora, comprehensive baseline comparisons, systematic error analysis, and enhanced technical rigor before it could meet publication standards for a venue of this caliber.

---

### Note · Authors · 2026-03-09

I have read and agree with the venue's withdrawal policy on behalf of myself and my co-authors.

---

### Meta-Review · Area_Chair_UBtr · 2026-01-05

**Summary:**

The reviewers found that the contributions of this submission were limited along multiple dimensions: in terms of evaluation (single corpus of 67 articles), in terms of methods investigated, in terms of analysis of the results. The writing was also seen as a bit light in terms of establishing methodological soundness and justification.
There was no rebuttal, and thus no interaction to back soundness.

**Reviewer Concerns:**

There was no rebuttal

**Reviewer Scores:**

The authors did not provide a rebuttal, thus the reviewers would not have changed their score.

---

### Decision · Program_Chairs · 2026-01-26

Reject